# The Role of Health in Households’ Balancing Act for Lifestyles Compatible with the Paris Agreement—Qualitative Results from Mannheim, Germany

**DOI:** 10.3390/ijerph17041297

**Published:** 2020-02-18

**Authors:** Alina Herrmann, Rainer Sauerborn, Maria Nilsson

**Affiliations:** 1Heidelberg Institute of Global Health (HIGH), Heidelberg University Hospital, 69120 Heidelberg, Germany; rainer.sauerborn@uni-heidelberg.de; 2Department of Epidemiology and Global Health, Umeå University, 90187 Umeå, Sweden; maria.nilsson@umu.se

**Keywords:** health co-benefits, climate change, health, sustainable lifestyle, equity, prevention

## Abstract

Household lifestyles are the main drivers of climate change. Climate change mitigation measures directed to households often have substantial health co-benefits. The European mixed-methods study HOPE (HOuseholds’ Preferences for reducing greenhouse gas emissions in four European high-income countries) investigates households’ preferences for reducing greenhouse gas emissions and particularly researches the role of information on health co-benefits in households’ decision making. The results presented in this study are derived from 18 qualitative interviews, conducted with a subsample of households from Mannheim, Germany. The in-depth interviews were transcribed verbatim, analyzed with a qualitative content analysis, supported by NVivo software. They showed that, in order to reduce their greenhouse gas emission in a way compatible with the 1.5 °C goal, households have to undertake a difficult balancing act, considering factors from the individual sphere, such as health co-benefits, as well as from the public sphere, such as (climate) policies. Shared responsibility and equity are important aspects of households. In conclusion, health is an important factor in households’ decision making. However, information policies about health co-benefits need to go along with structural policy measures, in order to support households effectively in the implementation of healthy and climate-friendly lifestyles, especially in sectors where behavior change is difficult, like the mobility sector.

## 1. Introduction

Anthropogenic climate change endangers the natural resources and sociocultural foundations of human civilization and human health [1,2]. This is because rapid temperature changes, extreme weather events, ocean acidification and sea-level rise can lead to aggravation of food insecurity, changed infectious disease patterns, social conflicts, migration and deaths [2,3,4]. Therefore, in the Paris Agreement, 187 countries have agreed to limit global warming to well below 2 °C, and possibly 1.5 °C [5,6]. In order to reach the 1.5 °C goal, greenhouse gas (GHG) emissions would need to decline globally by about 45% from 2010 to 2030 [7]. Households control up to 72% of global anthropogenic GHG emissions, when emissions are counted from a bottom-up perspective [8]. The houses people live in, the way they travel, the things they eat and how they recycle their waste, as well as all other forms of consumption determine the amount of GHG they emit. Thus, households are key actors in the transition to a low-carbon world [9,10,11,12,13]. A common measure to assess all GHG emissions a person or a household emits during one year is the carbon footprint [14]. The carbon footprint refers to carbon dioxide (CO_2_), but also considers other GHGs than CO_2_, such as methane (CH_4_) or nitrous oxide (N_2_O) and accounts for them as CO_2_ equivalents (CO_2_e). 

Many lifestyle features determining individual carbon footprints impact health [15]. Therefore, measures to reduce GHG emissions can also offer substantial health co-benefits. By definition one speaks of health co-benefits, if actions primarily targeting climate change mitigation, also promote health [2]. A common example of health co-benefits of political mitigation action is the reduction of air-pollution related morbidity and mortality, due to shifts from fossil fuels to renewable energies [16,17,18]. This kind of health co-benefit depends on political action and is not directly accessible to individual action of households. However, there are many so-called direct health co-benefits which are accessible to individuals depending on their own behavior, not necessarily being conditional on policies or other people’s action [19]. 

Such direct health co-benefits mainly occur in the sectors of mobility, food and housing. In the mobility sector, they mainly arise from active travel compared to car or motorcycle use. Active travel describes modes of transport, including physical activity, such as walking, biking or using public transport leading to the prevention of cardiovascular diseases, diabetes, some cancers and better mental health [20,21,22,23,24]. In the food sector, health co-benefits can arise from a shift to a more plant-based diet [25,26]. This is because livestock production is responsible for about 80% of global agricultural GHG emissions [27]. At the same time, lower intake of saturated fats from animal source and highly processed or red meat lowers the risks for cardiovascular diseases and bowel cancer [28,29,30]. Furthermore, the intake of highly processed food in general increases, e.g., cardiovascular health risks and goes along with high GHGs emissions from processing and packing [31,32,33]. In the housing sector, better insulation of houses can save energy and at the same time reduce cold- and heat-related morbidity and mortality [34,35,36].

As households can contribute greatly to climate change mitigation through their lifestyle choices and at the same time gain benefits for their personal health, the question arises if framing mitigation action in positive health terms, can motivate households to live more climate-friendly. It has been stated that health could be an efficient motivator to implement climate change mitigation action [37,38,39]. However, the empirical evidence that this holds true for households remains inconsistent to date. In a US sample, neither health nor economic co-benefit frames increased policy support or own behavioral intentions for climate change mitigation [40]. In contrast to that, Myers et al. showed that a public health frame aroused hope and support for climate change mitigation across a varying spectrum of climate change beliefs in another US sample [41]. So far, most studies investigating the effect of health framing did so with reference to avoiding negative health impacts of climate change or achieving indirect health co-benefits, such as the ones from reducing air pollution. 

To shed light in this debate, we carried out a study on “Households’ Preferences for reducing GHG emissions (HOPE)” in four mid-size cities of European high-income countries: France Germany, Norway and Sweden [19]. In line with the 1.5 °C goal, asking substantial reduction efforts especially from high-income countries, the study engaged households in a simulation game, in which they were asked to half their carbon footprint by 2030. We also examined the role of direct health co-benefits in households’ decision making on climate-friendly lifestyles. The study comprised quantitative and qualitative parts.

In this paper, we report the qualitative results of the German sample of the HOPE study in order to explain the German household’s decision making on their way to half their carbon footprint in line with 1.5° goal. A focus of this paper is to investigate the role of direct health-co benefits in relation to other factors determining households’ willingness to reduce their carbon footprint. In order to do so, we will also consider the quantitative findings from the HOPE study in the discussion section. 

The main finding is that health is an important factor in households’ decision making. However, information policies about health co-benefits need to go along with structural policy measures, in order to support households effectively in the implementation of healthy and climate-friendly lifestyles, especially in sectors where behavior change is difficult, like the mobility sector.

## 2. Materials and Methods 

### 2.1. The HOPE Study

The HOPE study investigated “Households’ Preferences for reducing GHG emissions in four European high-income countries (HOPE)” in four mid-size cities in France (Communauté du Pays d’Aix), Germany (Mannheim), Norway (Bergen) and Sweden (Umeå). The transdisciplinary study had a mixed-methods study design, including three sequential interactions with households, of which the first two elicited quantitative data, and the last one elicited qualitative data. A sequential explanatory mixed-methods design means that the qualitative data collection and analysis is based on the preceding quantitative findings in order to be able to complement and to explain them. In such a design, qualitative findings have equal scientific value and stand for themselves [42]. Furthermore, a policy analysis investigating household oriented-climate policies on a local, regional and national level was conducted [10]. 

A comprehensive study protocol with detailed information on objectives, materials and methods was published in 2017 [19]. Quantitative data collection was conducted from June 2016 to November 2016; qualitative data collection was carried out from February 2017 to September 2017.

### 2.2. Mixed-Method Design and Procedures

The first interaction was a quantitative questionnaire, which gathered socio-economic data and assessed households’ carbon footprints.

The second interaction was a simulation game. Participants were confronted with the task to reduce their carbon footprint by half. In order to do so, they could choose from up to 65 mitigation actions in the sectors “food”, “housing”, “mobility” and “other consumption”, which were presented to the respondent on action cards, which also contained information about CO_2_ reduction, costs/savings and possible health co-benefits. In the first round, participants could choose voluntarily which actions they would like to implement. If they had not reduced their footprint by 50% in the first round, they entered a second round, in which they were asked to continue to pick more mitigation actions, until they reached a 50% footprint reduction.

The third interaction was the qualitative in-depth interviews which were carried out on a subset of the quantitative study sample. It is this study component, which we present here. 

### 2.3. Study Participants and Qualitative Sampling 

In the quantitative study, 308 European households completed Interaction 1 and 2. We chose a purposeful sub-sample of these 308 households for qualitative interaction 3. In order to capture many different perspectives, we applied a maximum variation sampling [43]. Due to Interaction 1 and 2 we knew participants’ socio-economic characteristics and preferences in climate change mitigation well, so that we were able to select the households purposefully. On the one hand, we selected households, which differed in the following categories: Gender, age, education, income, nationality, housing status (renter, owner), housing type (apartment, single house), household location (urban, suburban, rural), household size, presence of children (<18y) in the household, whether or not the households had received the health information in Interaction 2, different levels of GHG reduction as result of the first and second round of the simulation game. Furthermore, we selected households, which covered certain extremes in these dimensions. Concretely, we selected households with the highest and lowest initial footprint, at least one climate-sceptic household, at least one household which was a climate policy or technology expert, at least one household who refused to reduce the footprint by 50% and at least one household who already reduced the footprint by 50% in the voluntary round. The sample size in each country was defined by the principle of saturation, which means that no more interviews were conducted, when the contents of the interviews reached a high grade of repetition and few or no new themes come up [44]. In the German sub-sample, which is presented in this paper, saturation was reached after 18 interviews. All German qualitative interviews were conducted in German by the first author. Quotes in this paper are translated. Original quotes can be obtained from the first author. 

### 2.4. Development of the Qualitative Interview Guide 

The interview guide was developed by an interdisciplinary research team with backgrounds in public health, medicine, political science, social science, journalism and engineering. There was at least one researcher from each country (France, Germany, Norway, Sweden). All researchers had been involved in designing the mixed-methods study, and most had been involved in the quantitative data collection. 

The interview guide was developed as a semi-structured interview guide, which means that it contained different sections with a list of open-ended and possible probing questions. On the one hand, it is important to have a standardized list of questions to make sure that aspects, which are important to answer the research question, are covered in the interview [45]. On the other hand, we handled the order of questions and the depth probing of certain aspects in a flexible way. This is important because in-depth interviews should be close to a natural flow of conversation, so that the interviewee feels free to share his/her thought and feelings [46]. The HOPE interview guide contained five sections, as summarized in Table 1. 

As Table 1 shows, the interview guide was closely related to the answers and results of the individual’s quantitative interview part. In Section 4, we applied the technique of invoking thoughts and views based on short narrative scenarios. The interview guide contained seven household scenarios, which put the household into a certain situation and asked how the participant would act (e.g., Imagine you would like to reduce your meat consumption, but your partner/family loves meat. How do you feel about this? What would you do? What would make it easier to act in accordance with your own values?) Moreover, it contained about eleven, partially country-specific policy scenarios (e.g., “Imagine that your government would promote sustainable food production and decrease subsidies for conventional food production. This could make food more expensive for you. What do you think about this? How would you feel about such a decision made by others?”). The scenarios often contained further probing questions. Depending on the course of the interview, the interviewers chose one or more household and one or more policy scenarios, which were suitable to discuss with the specific household. 

### 2.5. Qualitative Data Analysis 

Before the analysis, the interviews were transcribed verbatim according to the transcription scheme based on the one from Dresing and Pehl [47]. In the analysis, we combined an inductive and a deductive approach. We applied an inductive approach of qualitative content analysis in order to explain the German household’s decision making on their way to half their carbon footprint in general. Inductive approaches in qualitative research mean that a theory or theme(s) are generated from the data, also described as ”grounded” in the data [48,49]. We conducted the process of our inductive content analysis according to the concepts of Graneheim and Lundman. Their first unit of analysis is the meaning unit, which is then condensed in order to create a code. This means that we identified text passages, which related to one central meaning and marked those with a code, which reflected the condensed meaning of that text unit. In a further process of abstraction, those codes can be subsumed to (sub-) categories and later (sub-) themes. According to Graneheim and Lundman (sub-) categories subsume contents, which share a communality. While they are rather descriptive, and therefore, consider manifest content, themes are more interpretive, and therefore, express a summarized latent content of codes and categories [50]. 

In addition to this inductive method, we applied a deductive approach to specifically investigate the role of health co-benefits and explain the sectoral differences of the quantitative health findings. Deductive approaches are used, if there are pre-defined theories and initial concepts or variables that are to be further explored and corroborated with the qualitative data [51,52]. This is the case for health in the mixed-methods HOPE study. The hypothesis from the quantitative study (so the pre-defined theory) is that information on direct health co-benefits significantly increases households’ willingness to act climate-friendly, albeit only in the food and the housing, but not the mobility sector. Qualitative findings should now extend the quantitative ones and help to explain relationships between health and other variables [52]. Therefore, after having asked open-ended questions, also specific questions targeting the role of direct health co-benefits in households’ decision-making were introduced in the interview guide (see Table 1). In the analysis, Deductive analysis was accomplished by applying pre-defined subcategories to the interview material and adapt them if necessary [51]. We approached the material with a rough pre-definition of health subcategories along the lines of the sectors applied in the quantitative study. This means that we coded meaning units, which contained perceptions of health in the four pre-defined sectors of mobility, housing, food and other consumption. In the course of our analysis, we further adapted these predefined subcategories by adding one more general category called “General perceptions of health co-benefits”, because participants described health co-benefits which did not fit one of the sectors. Moreover, within each of the sectors, we further differentiated health perceptions as hindering, facilitating or neutral towards the implementation of mitigation action.

Finally, we sought to integrate the categories build in the inductive approach with the deductive category of “Health” on a higher level of abstraction, as will be described further in the results. 

In Germany, the analysis was mainly conducted by the first author. Three interviews were re-coded by a research assistant with a first version of the coding scheme. The coding scheme was then discussed and revised. The process to develop sub-categories, categories and the main theme from the codes was an iterative process, going back and forth during the process of analysis.

### 2.6. Assumptions on Health Co-Benefits

Prior to the quantitative study, the research team made conservative assumptions based on scientific literature about which mitigation actions had scientifically proven direct health co-benefits for households. Eleven health co-benefits and one health co-harm were identified for 12 out of 65 mitigation actions in the sectors of Food, Housing and Mobility, but not Other Consumption. The semi-quantitative assumptions about the size of the health effects were derived from scientific literature and based on an epidemiological measurement of disease-burden, called qualitative adjusted life years (QUALYs). Table 2 summarizes the identified health co-benefits and their estimated health effects, as presented to participants. 

A more detailed version of the table with explanations of the method, the size and nature of health effects can be found in the study protocol paper [19]. 

## 3. Results

Being asked to reduce one’s carbon footprint by 50% was a challenging scenario for all households in the simulation. During the course of the qualitative interview, participants brought up many different aspects, which we coded and grouped into categories and subcategories. Each category related either to the individual sphere or to the public sphere. All in all, the emerging core theme relating to all categories, was that it was a balancing act for households to consider all those different aspects to reach a more climate-friendly lifestyle. Another underlying theme catching important latent content from all categories and feeding into the core theme was that households saw climate change mitigation as a shared responsibility between all actors in the private and public spheres, which should be achieved globally and in a socially equitable way. Figure 1 gives an overview of the two themes and the categories. 

The two underlying themes (latent content) are set in the middle. Categories (manifest content) within the public and individual sphere are grouped around the themes. The health category is set in additional dashed lines, as it was explicitly introduced into the interview guide and analyzed deductively.

As one can see in Figure 1, households’ considerations about direct health co-benefits were part of the individual sphere of the balancing act. While the other categories emerged from the interview data inductively, the health category was partly brought up by participants themselves and—if not— partly probed by the interviewer (mixed inductive and deductive approach). Therefore, health is marked with additional dashed lines. 

In the following, we will first describe the manifest themes in the categories (3.1.), followed by the latent content in the themes (3.2.).

### 3.1. Categories

Some of the categories rather pertained to the individual sphere, such as health, knowledge and awareness, personal values, financial considerations or convenience, time and habits. (3.1.1.–3.1.5.). Other categories rather pertained public sphere, such as societal issues, the role of industry and the service sector and climate policies (3.1.5–3.1.8). Most of the categories contain both barriers and facilitators for implementing mitigation action. Table 3 illustrates the sub-categories leading to categories, sub-themes and finally themes. 

#### 3.1.1. Health

Some respondents considered a sustainable society and nature preservation in general is good for health with several perceived health co-benefits. In particular, nature was perceived to be a source for energy and health, while air pollution and other environmental poisons were perceived as health-damaging. Furthermore, some participants also associated a more sustainable society with benefits for mental health, for instance, because of slower, more mindful lifestyles or because they expected a better work-life-balance, due to societal transformations (e.g., basic income in green economy). 

Table 4 presents illustrative quotes of participants health perceptions. One can see that they perceived health co-benefits, but also co-harms of mitigation actions. Below the table, we will present the qualitative findings according to sector-specific health co-benefits (four subcategories) as tested in the quantitative study. 

**Health aspects in the food sector.** In the food sector most participants underlined that high quality food was important for their health and well-being. However, more climate-friendly food, in particular, organic, regional, seasonal, less-packaged and vegetarian food, was judged in differentiated ways. Some participants said that they particularly enjoyed regional and seasonal food. Some also perceived organic and less packaged food to be healthy, due to the avoidance of potentially harmful chemicals (e.g., fertilizer, antibiotics, hormones). Other participants found health benefits of organic food discussable and plastic packing to be more hygienic, lighter to carry, and therefore, healthier. 

Participants had differentiated views on meat consumption: Some participants deemed fully vegetarian diets to be unhealthy, because they found that they would miss out on important nutrients. Furthermore, two participants said that the vegetarian alternatives in their canteens would be unhealthy, due to high share of carbohydrates or high calorie content. Fish was explicitly mentioned by a few to be part of a healthy diet. Reduced meat consumption was perceived to be healthy by some, partly due to reduced fat intake. Several participants expressed their preference for regional and organically farmed meat, partly for reasons of animal welfare and climate, but also partly because of health reasons like less use of antibiotics and hormones.

**Health aspects in the mobility sector.** The far most important health co-benefit was perceived to be reaped in the mobility sector. Active mobility, such as walking and biking was viewed as both healthy and fun. Participants thought that active modes of transport would reduce stress, made them get some fresh air, made them integrate exercise in daily life, and that movement would lead to well-being. Furthermore, few mentioned that reduced air pollution and noise by the reduction of car use would also be healthier for all. At the same time, few participants pointed out that cold, bad weather, smog, risks of traffic injuries or being violated in dangerous areas were risks associated with walking or biking. In relation to air travel, it was sporadically mentioned that radiation was bad for health and safety concerns, due to terrorism could discourage from air travel. 

**Health aspects in the housing sector.** In the housing sector, health was less often perceived as a relevant issue than in other sectors. Some participants expressed that a good building structure with good materials could be beneficial for health. Sufficient room temperature was also seen to be important for health and well-being. However, some participants also perceived too much insulation to provoke mold and be worse for indoor air quality. Wood stoves, which were seen as beneficial for the climate, were at the same time described as harmful for health, due to air pollution by some. 

**Health aspects in other consumption.** In the sector other consumption, which comprised things like clothing, cosmetics, leisure time activities and choice of accommodation during holidays, health co-benefits were mostly encountered in the consumption of natural or organic products. Participants wanted to avoid harmful substances like chemicals in clothes or cosmetics, especially for children. Other mitigation actions which were related to health co-benefits sporadically were: Less consumption of television and digital device (e.g., better sleeping quality), active vacation (hiking/biking), to quit smoking. Furthermore, one participant regarded low carbon holidays like camping as potentially harmful for health in older age.

#### 3.1.2. Knowledge and Awareness

All participants could describe at least some impacts of climate change. The increase of extreme weather events was mentioned most often, followed by melting of glaciers and less snow and the effect of climate change on increasing migration. Aspects like sea-level rise, change in seasons and loss of biodiversity and other environmental degradation were mentioned less often. Many, but not all, participants expressed that knowing about the impacts of climate change caused concern and urged them to take actions. Yet some also felt undisturbed by climate change, because they either doubted anthropogenic climate change or thought that humans and nature could adapt. 

When talking about how to reduce GHG emissions, it became obvious that being aware of one’s own (un-)sustainable behavior and knowing what to do to mitigate climate change was seen as a facilitator for action. However, several participants described that they often acted against their better knowledge of what would be more sustainable, sometimes going along with a bad conscience. Some explicitly said that despite being convinced of the importance of sustainable behavior, they would not be able to change their lifestyle sufficiently without policy support. 

While missing knowledge on how to live sustainably was obviously seen as a barrier to climate action, another important barrier mentioned by virtually all participants was the complexity of sustainable choices. This complexity pertains to the following problems: First, participants found it difficult to assess the lifecycle emission of products and services. For instance, one participant doubted that insulating his house would really save emissions, if one would also consider the process of producing, transporting and disposing of the insulation material. Second, some found that there were measures, which were good for the climate, but bad for the environment. An example for this, as previously mentioned, were wind power plants, which would disfigure landscapes and endanger some bird species. Third, buying green electricity was seen as critical, because it was not transparent for the clients, if the electricity coming to the house was actually green or if the green electricity contract would really promote renewable energies. There were many other examples, in which participants articulated doubts or uncertainty about the effectiveness of the proposed climate change mitigation measures, possibly due to lack of information. These doubts and uncertainty, due to the complexity of the matter often made people rather stick to their current lifestyle instead of taking climate action.

#### 3.1.3. Personal Values

Participants also expressed values and internal attitudes that influenced their choices for reducing GHG emissions. Table 5 gives an overview of values that were rather facilitating or rather hindering participants of implementing climate action. 

As an example of a facilitating value, almost all participants named nature and environmental conservation as an important value to take action. Many also described that they wanted to act, because they felt connected to nature. The majority of participants considered our current societal living standards to be high and partly were ready to refrain from some of these comforts consciously. Many participants also wanted to appraise the real esteem of goods and rather buy good products once and then use them for a long time. Mainly in the food sector, but also in other sectors, people perceived that sustainable choices went along with a high quality of life. Furthermore, values like animal welfare or fair labor and trade conditions played an important role in choosing more sustainable options in the sectors food and other consumption. 

Although values connected with nature could be facilitators for climate action, for instance, in the food sector, wanting to experience nature could also be a barrier to sustainable behavior, especially in the mobility sector. Table 6 shows how wanting to experience nature can lead to carbon-intensive but also carbon saving behavior. In general, many facilitating values could be seen in the food sector, while hindering values were rather seen in the mobility sector. 

Some participants also expressed that it was inherent to humankind to strive for progress and prosperity, so that self-chosen sufficiency and simpler lifestyles were seen as unrealistic. However, those participants did not argue against sustainable lifestyles, but either reasoned that policy makers would be the ones who would need to make restrictions for everyone or that the system would need to act more sustainable as a whole. 

#### 3.1.4. Convenience, Time, and Habits

The feasibility to implement mitigation actions was an important argument for all participants. Besides financial considerations (see below) convenience, time and the readiness to change habits or not were the most important aspects mentioned, either as barriers or facilitators.

One important barrier for implementing mitigation actions was that the climate-friendlier options were inconvenient or did not meet the participants preference or taste. For instance, greater comfort and flexibility of the private car compared to public transport or car-sharing, was often mentioned, especially by families. Sometimes, climate-friendly alternatives were simply perceived to be impossible (e.g., no public transport options available). A similar aspect was the aspect of climate-friendly options to be more time-consuming. This did not only pertain to the mobility sector (flights faster than other options, car faster than public transport), but also to food and other consumption. Buying regional, local organic or sustainably produced products, growing and cooking own food, or repairing products instead of buying new ones was seen to be as too time-consuming and partly also as ineffective. 

All participants expressed that another barrier to live more sustainably was the difficulty in changing habits. Many participants declared that missing will or self-efficacy were hurdles to change their behavior. Some argued that changing was difficult, due to a generally stressful life, where often other things seemed more urgent than sustainable behavior. Moreover, participants felt that certain lifestyle aspects were engraved in their personality, due to important lifestyle events. For instance, a male, older participant associated freedom with his car, because in his youth, this is what his first own car meant to him. A lady who had been raised in the former communist east of Germany recounted that her family was not free to choose an apartment, but faced strict spacious restrictions. Therefore, she nowadays enjoyed her big house and their living standard and did not want to miss it. Another important aspect under this category was that most participants articulated that they were ready to change some habits, but that a radical lifestyle change was unwanted. 

While convenience, time and habits were more often mentioned as barriers, they could also be facilitators. This was obviously the case, if sustainable alternatives were perceived as more convenient, or faster. For instance, some participants perceived the bike to be the quickest alternative on some routes or found train or public transport use to be more relaxed than driving the car. Concerning habits, participants described it as facilitating, if they already had implemented sustainable routines. To change habits, participants considered that it needed openness to change and time to investigate how the change could be done (e.g., What are protein alternatives to meat?). Furthermore, they said it needed time to implement the changes, either step-by-step (e.g., using the bike for more and more of daily routes) or by getting used to self-applied rules (e.g., no coffee-to-go without a reusable cup). 

#### 3.1.5. Financial Considerations

Financial Considerations were another theme of the balancing act. Generally, the costs of mitigation actions were perceived as a barrier. Common examples of this can be found in the insulation and energy investments in the housing sector, but also all other sectors. Especially in the housing sector, the aspect of complexity, in particular, knowing at what point of time investments would pay off, was a barrier for implementation. Other examples were expensive electric cars and far-distance-train rides in the mobility sector and expensive organic food or expensive organic clothing or hotels in the sectors food and other consumption. Some older participants pointed out that greater investments would not pay off in older age, and many participants found that socially deprived had fewer chances to act sustainably, due to their low income. However, most participants also considered facilitating aspects despite higher costs. First, most participants said that they were willing and able to pay higher prices for sustainable alternatives, if other benefits in terms of CO_2_ reduction or better product/service quality were clear. Second, many emphasized that small costs, such as those from organic products, hindered less than bigger ones, like investments in housing. Third, having money was seen as a facilitating factor by some. 

#### 3.1.6. Societal Issues (Public Sphere)

Under the topic of societal issues, we subsume broad statements on politics, the economic systems and other societal issues. Statements which refer more specifically to the industry and service sector or to concrete policies, are treated independently there. As all participants pointed out that climate change mitigation was a shared responsibility, allocations of responsibilities to governments, industry/service sector and individuals are subsumed under this theme (see 3.2). All participants identified barriers and facilitators for implementing mitigation actions in society. Table 7 illustrates these barriers and facilitators, as well as the analysis process from identifying a meaning unit to deriving a code and subcategory in the category “societal issues”.

Most participants expressed that our modern society was to some extent a barrier to sustainable behavior, due to societal norms evoked by technical innovations, globalization and market-economy. For instance, the availability of technical innovations like one-time-use packaging, consumer electronics or planes would lead to less sustainable lifestyles automatically. Some emphasized that the current principles of the market economy, e.g., the need to sell new products to create growth, would oppose sustainable principles. Furthermore, some pointed out that in a globalized world it has become normal and/or necessary for individuals and goods to travel long distances, often by plane. Most participants also reported that the current interplay of governments and the industry/service sector produced hindering financial incentives, such as flight being cheaper than train rides. Many participants also pointed out that climate change was only one of many relevant public issues and that other things, such as providing enough jobs also needed to be considered. Missing infrastructure was also named as a hindering factor. 

#### 3.1.7. Industry and Service Sector (Public Sphere)

The aspect, which was most often highlighted by participants on this area, was that industry and service sector would need to offer good quality and affordable, sustainable products and services, so that household could actually buy and use them. So far, most participants thought that there were too little sustainable alternatives. For instance, food offered in supermarkets would often be only available in plastic packaging, and the offer of organic or regional food was not sufficient. Another example was the quality and price of electric cars, which was deemed as insufficient compared to combustion engine cars. Some participants also uttered that there were not enough possibilities to rent or buy passive-standard houses and apartments. Some participants also criticized that local production was not considered enough by industry. While participants, on the one hand, expected prices to be affordable, they also wanted the price to reflect the real value of products or services, so that, for instance, farmers could earn a decent salary. Many participants also claimed that a great barrier for sustainability was that industry and service sector were not intrinsically interested in sustainability, so that even some sustainability campaigns would mainly be green-washing without bringing about a real change. 

#### 3.1.8. [Climate] Policies (Public Sphere)

Many participants expressed that they thought it to be difficult to meet the European and German emission reduction targets with current policies. While most participants at least once indicated that they preferred soft policy measures, they also acknowledged that stricter policy regulations are needed. One reason to prefer soft policy measures, such as information campaigns and incentives was that participants asked for respect for individual freedom of choice. Some pointed out that significantly higher and unavoidable costs could lead to higher social inequity, and in this case, would be unacceptable. However, most thought higher costs for products and services with higher CO_2_-emissions acceptable, if this would not touch basic needs and still leave room for maneuver. For instance, several participants described that they would accept significantly higher costs for flying, because then they would fly less (e.g., every three years instead of each year) but would not need to give up flying as a whole. Furthermore, most participants declared that if strict policy regulations would come and be valid for everyone, one could adapt. Many participants described existing policies as dysfunctional, for instance, funding programs in the housing sector, due to difficult bureaucracy.

Building knowledge and awareness in all areas of society, starting from early childhood education over public media to vocational trainings, was seen as one key policy measure. Furthermore, the provision of sustainable infrastructure from bike lanes to better and more affordable public transport was deemed crucial by many. The sector, in which participants were most favorable towards soft policy measures, but also strict ones like taxations and regulations, was the food sector.

### 3.2. Themes

#### 3.2.1. The Balancing Act for a Sustainable Lifestyle

The variety of manifest contents described in the categories above emphasize that participants did not think one-dimensionally, but rather needed to undertake a balancing act in their struggle for a climate-friendly lifestyle. Although the simulation was solely directed at climate change mitigation, participants often strived more holistically for sustainability. This became particularly clear, when looking at participants’ personal values (see 3.1.3). Almost all participants mentioned nature and environment conservations as a whole, including care for e.g., saving of drinking water or clean environments without pollution. Furthermore, many deemed animal welfare to be an essential aspect in the food sector, and others considered fair labor conditions and fair trade to be important. Therefore, we labeled the main theme “balancing act for a *sustainable* lifestyle”.

But how to balance? Households described mostly implicitly how they approached the difficult balancing act for reducing GHG emissions and live more sustainably. Table 8 illustrates households approaches on how to balance. 

First, all participants tried to take balanced and proportionate action (see Table 8). That meant that they, on the one hand, weighted barriers and facilitators from the different categories against each other. On the other hand, they also weighted the amount of carbon savings the actions would yield against those barriers and facilitators. Many participants also described how they made exceptions for carbon-intensive behaviors in special life situations, such as visiting a child which works on another continent via far-distance flight, although not flying otherwise. Second, respondents described that it might be easier to take opportunities for GHG savings at turning points in life, such as moving to a smaller apartment for retirement. Third, as part of the balancing act, participants set their actions into comparison: They either compared their GHG-intensive behavior to other areas in their life, where they were saving GHGs. Similarly, they compared their own GHG-intensive behavior to other people, who emitted even more GHGs. This often occurred, when people were not ready to implement a proposed action. One could see this strategy as a discounting of GHG-intensive behavior in order to reduce cognitive dissonance. Therefore, this last strategy on how to balance was rather a barrier to implementing ambitious mitigation action. However, the fact that participants put themselves into comparison with others already leads to the second important underlying theme, namely, shared responsibility and equity. 

#### 3.2.2. Shared Responsibility and Equity

While the HOPE simulation game was mainly directed on individual climate change mitigation action, all participants emphasized that climate change mitigation was a shared responsibility and that the principle of equity needed to be considered. Participants felt that society as a whole needed to act. While many different stakeholders were named, including agriculture, research and education, almost all participants referred to governments, individuals and the industry/service sector. 

Governments were considered important for setting standards and organizing international collaboration. Most participants emphasized that in the end, only global action would be effective. Some also referred to government’s responsibility as a role model. In this sense, many felt that trust, transparency and authenticity of governmental action were important for them to act. This included that laws, regulation and policies set by governments should be actually enforced for their goals to be effectively reached. Most participants criticized that lobbying by special economic interests would hinder effective governmental action, and some found that governments would generally act too little on climate change. 

The industry and service sector where seen to have a strong negative influence on climate change mitigation. Participants pointed out that they had big opportunities to reduce GHG emission. Those were seen in the development of sustainable goods and services, the application of technical solutions to reduce GHG emission in production or in the reduction of business travel. However, some households felt that interest in profits would hinder this sector in taking action. 

Individual action was generally seen to be important as part of collective action. This meant that households acknowledged and generally accepted their individual responsibility. However, many had experienced through the simulation game, how hard it was to reduce their carbon footprint by half in order to reach the 1,5° goal. Therefore, many participants pointed out that they only had influence within boundaries, and therefore, expected collective action. Participant 20222, a student living in Mannheim city center said:
*“*20222: *It was nice to see, that I can do something myself. […] I try to pay more attention now. […]* Interviewer: *Who do you think responsible for climate change mitigation? […]* 20222: *All of us. Well, also big enterprises and companies, […] politicians, famous people […] because in the end they can move something. I think it has to go hand in hand, there must be a change, that goes all the way through.”*(*20222, male, 24 years*)

Finally, this also leads some respondents to express their wish for more equity within climate change mitigation action. Households asked for inter-individual, intersectoral (individual/industry/public…) and international equity. Inter-individually, many participants with under-average carbon footprints criticized that they had to reduce by 50% as well. They felt that reducing a footprint that was already small was harder than to reduce a bigger footprint. However, households with higher footprints seldom found it easier to reduce their footprint. They compared their emission production to other individuals or sectors as well, for instance, they compared the emissions in the household sector to emission in the industry. Table 9 illustrates these dilemmas. 

Some participants explicitly stated that rich households should start to reduce their living standards and emissions. These types of statements illustrate that households had strong feelings about equity, which they mostly expressed by comparison to other people living in Germany. However, some participants also compared high living standards in Germany to lower living standards in other countries. They usually expressed that people in other countries should also be able to raise their living standards, few questioned their own higher living standards. On the broader international level, many participants also claimed that all countries needed to act, because the action of single countries would disadvantage those countries and not lead to effective climate change mitigation globally. 

## 4. Discussion

### 4.1. Discussion of Methods

This qualitative part of the mixed-methods HOPE study is important to reveal insights about households’ reason and reasoning, why they chose their GHG reduction options and to identify barriers and facilitators for this. It was a strength of the qualitative study that we were able to purposefully select a maximum variation sample with the detailed information we had about the participants from the preceding interactions. According to Marshall purposeful sampling aims at selecting a sample, which will give the richest information to answer the research question [44] When wanting to discover common, and therefore, important patterns and themes across a great variety of participants a maximum variation sampling is most suitable [43].

A limitation of this study is that it is limited to participants from Mannheim, Germany. However, qualitative studies per se are not aiming to produce representative samples, as sample size are too small. However, by reaching a strong internal validity trough in-depth research, results from qualitative studies are transferable to similar settings [53]. In this case, these are possibly other urban and suburban settings. As 76% of Germans live in urban and suburban settings [54], and cities are important players in climate change mitigation [55] the urban settings seems justified. 

As the role of the health argument in household’s decision-making was one of the study objectives, it was analyzed in a deductive approach, while the other categories were derived inductively from the data. While qualitative methods have been used more widely in recent years, a combination of deductive and inductive approaches, have become more common. For instance, Gläser and Laudel explicitly suggest a combination of the inductive and deductive building of categories for the analysis of expert interviews [46]. Likewise, Pope et al. explain how in health research a thematic framework can be built from a priori categories derived from the research objectives (deductive) and from new issues raised by respondents (inductive) [56]. 

### 4.2. Discussion of Health Results

The QUANTITATIVE part of the HOPE study showed that additional information about direct health co-benefits increased households stated mean willingness to implement climate change mitigation actions in the food and housing sector, but not in the mobility sector [57]. 

A first possible explanation offered by the qualitative findings presented here is that participants’ perceptions about health co-benefits and health co-harms in the food and housing sector somewhat differed from the information we provided in the quantitative study, while in the mobility sector participants’ perceptions overlapped with the information provided (see below). 

In the food sector, we informed participants that a greater share of vegetarian products was better for health. This information was particularly based on the reduced risks of heart disease and some cancers. It was given to households in a simplified semi-quantitative way (green symbols on action cards) [19]. In the qualitative interviews, only a few participants perceived reduced meat consumption to be healthier. If at all, they associated it with less fat intake, for instance, in pork. None of the participants associated a lower cancer risk with lower meat intake, and few thought an entirely vegetarian diet might lead to some nutrient deficiencies. In fact, the positive health effect of a greater share of a vegetarian diet on cardiovascular diseases is based on the reduced intake of animal-based saturated fats and depends on what meat is substituted with [58]. If animal-based saturated fats are substituted by plant-based polyunsaturated fats, this reduces the risk for cardiovascular diseases [28], whereas, replacement with carbohydrates or monounsaturated fats has no or uncertain benefits [59]. Evidence of colorectal cancer risk reduction, due to reduced consumption of processed meat is quite well established; evidence is a bit less strong for red meat [29,30,60]. All in all, in a simplified way, it is justifiable to label more vegetarian diets to be healthier. Having households’ perceptions from the qualitative interviews in mind, this information might have been new to participants, and therefore, could explain why the group receiving this information was more willing to implement food options and reduce more emissions in the food sector compared to participants who did not. 

A similar case can be made for the housing sector, in which households saw health co-benefits and co-harms in insulation materials, while the scientific literature usually describes co-benefits, due to more temperate houses/apartments and reduced heat-/cold-related mortality and morbidity [35,36]. However, one also has to consider that these studies are based on data from the United Kingdom, while building structure in Germany is different. Furthermore, it will depend greatly on the current status of the particular households’ insulation, if increasing insulation will have additional health co-benefits. While we only considered health co-harms of very low room temperature, participants also brought in risk perceptions about mold, flammable and toxic insulation materials and their possible later health co-harms at disposal. In summary again, our information about health effects was more positive than households’ overall perceptions, so that the positive information might have made the difference. 

In contrast to the food and the housing sectors, households’ perceptions in the mobility sector matched findings in the literature (see introduction). This meant that households mostly perceived active modes of transport to be healthy and also enjoyed these modes of transport. Some individuals also mentioned the negative aspects of traffic accidents or violations. Willingness to implement mobility actions might not have changed with the reception of health information in the quantitative study, because knowledge of the health co-benefits seemed to be very prevalent in the whole sample. 

Both, in the food and in the “other consumption” sectors, some households explained that they would prefer organic products (food, textiles, toys etc.) in order to avoid harmful substances, such as pesticides, fertilizers, antibiotics or other chemicals. While some research considers such “egoistic” health reasons to be the main motivator to buy organic food [61], a representative study from Germany showed that “altruistic” reasons, such as environmental and animal welfare were the strongest motivators to do so [62]. This fits our findings, as only a few people mentioned health as a main motivator to buy organic food or other products, while many brought up other values associated with sustainability. As evidence of the health effects of organic vs. conventional food and other products remains conflicting [63,64], we did not consider these aspects in the quantitative study. 

When considering the broader findings from both the qualitative results on the “balancing act for sustainable behavior” and quantitative HOPE results about the general preference of households, another explanation for the different effect of health information in the food and the mobility sector seems to fit: The food sector was generally the sector, where people were most willing to change their behavior, while the mobility sector was the one they were least willing to do so [65,66] In the quantitative data no relevant differences in between countries were found [57,65].

This quantitative finding responded well to the qualitative findings of the balancing act, because we found fewer barriers for implementation of food options than for the implementation of mobility options. This is particularly true for moderate food actions, such as eating 30% less meat, 30% more organic/regional/seasonal etc. [65]. Eating moderately less meat or moderately more organic were seen to be convenient to integrate into normal routines, e.g., because such products were available at supermarkets. Furthermore, at least moderate changes were associated with affordable costs and many facilitating personal values were tied to mitigation action in the food sector. Extensive changes in food consumption were associated with greater barriers, and therefore, less popular (e.g., inconvenience of buying all products at local farmer’s market, high prices of buying organic products only etc.). Before this background of rather facilitating factors in the balancing act, the additional information about health co-benefits could have made a difference in the food sector, because people also felt capable of acting towards this kind of healthier behavior. It has been shown that information strategies are most effective to promote behavior change in areas were the target behavior is convenient and faces little structural barriers [67]. 

In contrast, the transport sector was the one where people were least willing to change their behavior (Sköld et al., 2018). This was also reflected in the qualitative interviews, which showed that in this sector, people encountered many barriers for behavior change, such as time constraints, infrastructural short-comings, and needing to change habits. Additional health information might not have made a difference here, because people did not feel capable of acting towards the suggested healthier behavior. Other researchers, therefore, recommend strict policy measures (infrastructure, regulations, taxation, etc.) rather than informational strategies to promote pro-environmental behavior in areas were households are facing many barriers for implementation [68]. However, Gifford et al. also conclude that informational strategies might help to create acceptance for such strict policy measures. 

### 4.3. Discussion of Results of the Balancing Act for Sustainable Behavior as Shared Responsibility

Previous research has identified barriers and facilitators for climate-friendly behaviors, some of the psychological, some more structural ones [69,70,71,72]. While this research mainly focused on current behavior without particular reduction targets, the HOPE study put participants into a scenario, in which they were asked to reduce their carbon footprint by 50% by 2030 in line with the ambitious 1.5 °C goal of the Paris agreement. Within this scenario, we found many similarities and some differences to previous findings.

Lorenzoni et al. also divided barriers for implementation into an individual and social (in our study: Public) sphere. Examples for barriers similar to the ones found in our study were lack of knowledge, insufficient infrastructure or insufficient offer of sustainable goods and services [73]. Furthermore, these authors highlighted the fact that individuals closely took a look at actions taken by other individuals and other sectors, like government, businesses and industry and compare them to their own actions [68]. Finding that others did not act, was a great barrier to mitigation behavior in both studies. This corroborates that our subtheme “shared responsibility and equity” is an important barrier on the way to sustainable lifestyles. Our finding was all the more robust, as respondents mentioned it across many of our categories.

Compared to other studies, the uncertainty around if and how climate change will impact society and whether we need to act on it, was not much discussed by participants in our sample. This might be due to the study design, which put participants in a scenario in which they had to reduce their footprint, disregarding their general beliefs on climate change. Furthermore, it might be due to a selection bias, as the mixed-method study as a whole was rather time-intensive, so that people fully rejecting climate change might not have signed-up for participation. On the other hand, we had at least some climate sceptics in our sample. Those usually still could identify with the aim to live more sustainably, rather focusing on components of sustainable lifestyles, which do not directly relate to climate change, such as regional creation of values, high quality products or fair labor. 

The core theme, saying that households undertake a balancing act to reach a sustainable lifestyle should be discussed before the background of the pre-set goal of the simulation game, which was to reduce household emissions in line with the 1.5 °C goal. The interesting point here is the appraisal of climate policies in our study compared to others. On the one hand, we found, in line with previous studies [70] that many participants preferred soft policy measures, such as information campaigns and incentives. On the other hand, having experienced how hard it was to reduce emissions in accord with the Paris agreement, participants saw the necessity for tougher and better-enforced policies, such as taxes, regulations and rigorous implementation of a more climate-friendly infrastructure. Most of them also find these kinds of strict policies acceptable, if they apply to “all”. All, again, is pointing to the perceived need of equitable action between individuals, sectors and nations. An important aspect in this regard is that studies in the health sector have shown that people with lower socio-economic status benefit less from soft policy measures, such as counseling for better nutrition, than people with high socio-economic status. On the other hand, regulative policy measures, such as taxation of unhealthy food items, promote health equity between different socio-economic groups [73]. It is likely that this also holds true for climate mitigation actions and their health co-benefits. 

## 5. Conclusions

In sum, households consider many different aspects from the individual and from the public sphere, when they make choices about climate change mitigation actions compatible with the 1,5° goal set out by the Paris Agreement. It was a balancing act for them to consider a range of aspects reaching from personal values, knowledge and convenience to public aspects, such as the economic system and climate policies. It was important to all households that their own actions within this balancing act were embedded in collective action, which acknowledged a shared responsibility and principles of equity. 

Health, was one of the aspects households did consider in the balancing act. While the quantitative HOPE results showed that information about direct health co-benefits could increase households’ willingness to act on climate change mitigation at least in the food and housing sector, our qualitative findings suggest more specific conclusions: First, households have the need to be informed better about the specific health co-benefits of reduced consumption from animal products. In order to reap the best health outcomes, they also need to be informed about what to substitute them with. Such knowledge about healthy and sustainable diets should best be on all levels, starting from childhood education. Second, especially in sectors were households perceive many barriers to implement mitigation action, such as the mobility sector, only informing about health co-benefits is unlikely to yield substantial behavior change. More stringent climate policies, such as measures favoring active transport and discouraging individual car use, are needed. Being confronted with the aim to cut their emissions by half many participants acknowledged that strict policy measures are needed. 

Therefore, a multi-faceted policy approach with information strategies and incentives, but also taxations, regulations and infrastructural changes, is needed to support households in adopting more healthy and sustainable lifestyles effectively. Our results suggest that acceptance for such more stringent climate policies can be improved, if households understand the great challenge to reach the 1,5 °C goal and if they feel that policies respect principles of equity. Furthermore, informing about health co-benefits can increase the willingness of households to act, especially in sectors where there is little structural barrier for behavior change, such as the food sector. However, information policies about health co-benefits need to go along with structural policy measures, in order to support households effectively in the implementation of healthy and climate-friendly lifestyles, especially in sectors where behavior change is difficult, like the mobility sector.

## Figures and Tables

**Figure 1 ijerph-17-01297-f001:**
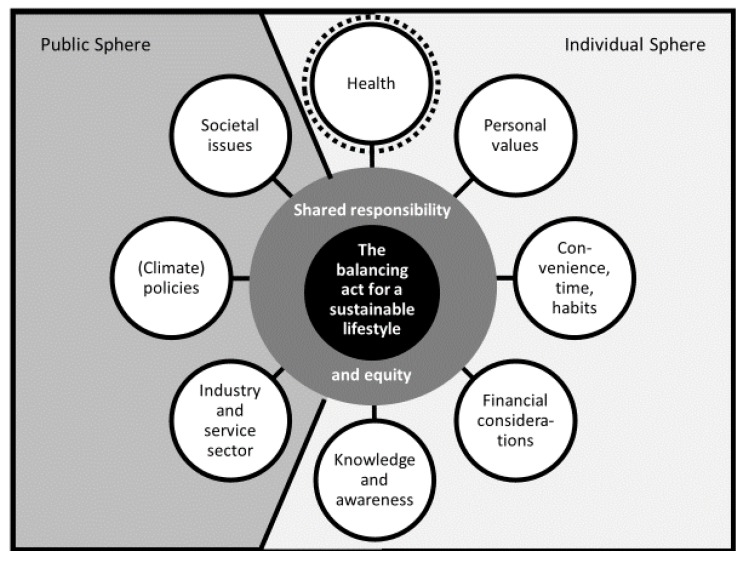
Themes and categories from the qualitative interviews.

**Table 1 ijerph-17-01297-t001:** Contents of the semi-structured interview guide.

No.	Sections	Contents
1	Introduction	Introduction of the interviewer, recapitulation of study procedures, the brief information of the objectives and procedures of the interview, how data would be used, handled and protected, ethical permit, confirmation of informed consent, possibility to answer the interviewee’s questions, appreciation for participation
2	Warm-up questions	Presentation of footprint results from interaction 1 + 2, first thoughts of the interviewee about this, perceptions of/experiences with climate change in general
3	Climate change mitigation actions	Presentation of mitigation actions which were chosen or not chosen by the household, questions relating to the reasons for household’s choices, questions about the importance of CO_2_e-reduction/costs/health in decision-making, questions on health co-benefits, questions on climate change mitigation in general (e.g., responsibilities, future visions)
4	Barriers and facilitators for change	Perceptions about the aim to reduce own footprint by 50%, barriers and facilitators for major changes, confrontation with pre-defined future change scenarios for households and policy (see text below table), household-specific questions
5	Closing questions	What enables/prevents change, the final message to policy makers, open questions to catch aspects which have not been treated yet

**Table 2 ijerph-17-01297-t002:** Assumptions of health co-benefits for the quantitative part of the HOPE study.

Sector	Mitigation Action	Health Effect
Food (and Recycling)	Eat 30% more vegetarian food (less meat and fish).	+++
	Eat 60% more vegetarian food (less meat and fish).	+++
	Become a vegetarian (stop eating meat and fish).	+++
	Gradually give up on ready-made meals (e.g., frozen pizza, canned soups, frozen lasagna).	++
Housing	Insulate your roof/attic.	+
	Insulate your walls.	+
	Improve your windows (increase glazing of your windows).	+
	Lower in-house temperature by 3 °C	-
Mobility	Shift significantly (more than 30%) from car to public transport (bus, tramway, metro, train).	++
	Shift to non-motorized modes of transport (walk, bike) instead of public transport.	++
	Decrease your travel with cars, public transport and other motorized vehicles by 30%.	++
	Give up your car(s) and other motorized vehicle(s)	++

Gains of QUALY on population level: <1 month QALY = +; 1–3 months QALY = ++; > 3 months QALY = +++; Loss of QUALY on population level: <1 month QALY = -. Source: Own project material, Briefing Sheet 4, HOPE-project www.hope-project.net.

**Table 3 ijerph-17-01297-t003:** Overview of subcategories, categories and overarching categories leading to themes.

Subcategory	Category	Overarching Categories	Underlying Theme	Core Theme
Facilitators in Society	Societal issues	Public Sphere	Shared responsibility and equity	The balancing act for a sustainable lifestyle
Barriers in Society				
Sufficient, affordable offers facilitate	Industry and service sector			
Little interest and insufficient offers hinder				
Soft policy measures preferred	(Climate) Polices			
Strict policy measures needed				
Current policies insufficient				
Dysfunctional policies hinder				
Concrete policy suggestions				
Knowledge and awareness facilitate	Knowledge and Awareness	Individual Sphere		
Complexity and missing knowledge hinder				
Knowl. and A. of climate change impacts				
Facilitating Values	Personal Values			
Hindering Values				
Convenience, time, sustainable habits and openness to change facilitate	Convenience, time, habits			
Convenience, time, habits hinder				
Savings of mitigation actions facilitate	Financial considerations			
Costs of mitigation actions hinder				
Facilitating considerations in spite of costs				
General attitudes on health co-benefits	Health			
Food and health				
Housing and health				
Mobility and health				
Consumption and health				

**Table 4 ijerph-17-01297-t004:** Illustrative quotes on health perceptions of climate change mitigation actions.

Illustrative Quotes	Code	Subcategory
*“We generally don’t eat a lot of meat […], once a week. But we eat cold meat in the morning. If I look at it from the nutritional perspective, I think we don’t eat enough meat. For instance, we try to eat low carb in the evening, in which case one should eat more meat and fish. […] I rather tend to iron deficiency […]. I find this difficult, very difficult… then I rather tend to look at the quality of the meat, but that is really expensive.” (20204, female, 30 years)*	Living without meat = unhealthy	Food and health
*“I like to ride the bike […] That’s movement and it reduces stress, it is exercise and exercise is good for everyone.” (20101, female, 43 years)*	Active transport healthy and fun	Mobility and health
*“There are insulation materials that are considered as critical for health reasons. In this case, I would definitely refrain from them, because health for instance would be more important to me than the assumedly reduced carbon footprint.” (20244, female, 40 years)*	Insulation material can harm health	Housing and heath
*“Meanwhile I often look for harmful substances in body care products, cosmetics, especially for the children. I also try this for clothing, but it doesn’t always work.” (20191, female, 42 years)*	Harmful substances in consumer products	Other consumption and health

Code of participant in parenthesis.

**Table 5 ijerph-17-01297-t005:** Codes in the two subcategories of personal values.

Codes	Subcategories	Category
Nature and environment conservation	Facilitating values	Personal Values
Feeling connected with nature		
Sufficiency and questioning our standards		
Care for future generations, being a role model		
Culture of appraising the real esteem of goods		
Perceiving sustainable behavior as high quality of life		
Animal welfare		
Fair labor and fair trade		
Pursuing something meaningful		
Independence from public infrastructure		
Desire for freedom and adventure	Hindering Values	
Keeping up relationships		
Considering all household members’ preferences		
Striving for prosperity		
Cultural experience		
Nature experience		
Education and career		

**Table 6 ijerph-17-01297-t006:** Illustrative quotes of some hindering or facilitating values in the food and mobility sector.

Illustrative Quotes	Codes	Subcategories
*“I like to be outside; I am a biologist, I have a relation to that. I like it, if you consume things that you have had in your own hand, that you have raised and that you can also eat. At this moment you have a different relation to the foodstuffs and a totally different respect. You don’t have that, if you buy it wrapped in plastic at Aldi’s. And that is why we do some home gardening.” (20101, female 43 years)*	Feeling connected with nature	Facilitating values (examples from the food sector)
*So for one thing I want to support organic farming, because I want to know what the food contains that I eat. If I buy milk or cheese, I want that the animals were kept properly and what is also important for me, is that the people, who are part of the production, earn proper money. [...] And therefore I find it ok, [...] if such food has its price.” (20186, female, 48 years)*	Animal welfare	
	Fair labor and fair trade	
*R: “First of all I am traveling through nature. Of course, this is also inconsistent, because one destroys much through air travel… Well, the experience, the life in nature and contact to other people, this is important to me.” (20226, male, 48 years)*	Nature experience	Hindering values (examples from the mobility sector)
*At the moment it is very important to have a semester abroad in your CV. I think in the companies you think: Hey, this guy is motivated, he wants to learn, he is flexible, he has been to the US for a year. It sounds better, then saying: Oh well, yes, this guy is organic, he is climate-friendly, he decided to stay at home and not pollute the air.” (20106, male, 44 years)*	Education and career	
*„Well, that question was problematic, because we flew to Japan twice in 2015 and to Hong Kong once in 2016. But the reason was to visit our daughter […] It was no flight “for fun” […] Instead, it was a necessity due to our family situation.” (20207, female, 64 years)*	Keeping up relations	

Code of the participant is in parenthesis.

**Table 7 ijerph-17-01297-t007:** Quotes, codes and sub-categories for the category “Societal Issues”.

Illustrative Quotes	Code	Subcategory
*Back then I thought it looks weird if I get there by bike. But by now I think this humbug. So there has been a change in society: It is seen as something positive to get somewhere by bike. (20106, male, 46 years)*	Change of societal values towards sustainability	Facilitators in society
*I absolutely support promotion of solar technologies and alternatives energy supplies“ (20231, male, 64 years)*	“Energiewende” (energy transition)	
*One must use the internet […]. All the public transport providers have a helpdesk […], where you get all routes and timetables. Thus, in the end it wasn’t a problem for me to get rid of my car. (20226, male, 48 years)*	Technical innovations facilitate	
*Our whole capitalistic system is not intended to reduce something […]. There are newly developed planes, that fly non-stop from London to California […] in 5 hours I think. Well, then the mistake is to make everything faster and bigger and higher and further and more expensive. And how high are the CO_2_ emission of that? One cannot tell me to buy A +++ appliances and at the same tome allow such plains to be developed! (20130, female, 71 years)*	Modern society traits: Technical innovations, globalization, market economy	Barriers in society
*And if there are options to use the train, then it is so expensive, that it is simply unaffordable. You just take the ICE train to Rome, you will be poor afterwards, no matter if you take first or second class, you really will be poor. (20118, female, 78 years)*	Wrong financial incentives	
*The thing is […] if it comes to* [loosing] *jobs […], then a lot of people are out of it. (20240, male, 38 years)*	Climate Change only one of many relevant public issues	
*If the bus only drives maybe three times a day, then one is quite restricted and can’t avoid to use the car. (20186, female, 48 years)*	Missing infrastructure	

Code of participant in parenthesis.

**Table 8 ijerph-17-01297-t008:** Illustrative quotes on how participants approached the balancing act.

Illustrative Quote	Code	Category
*If I can save a substantial amount of CO_2_ for 1000 Euro, then I do it of course […]. But there needs to be a proper relation* [btw. Costs and CO_2_ savings]. *(20230, male, 63 years)”*	Taking balanced, proportionate action	How to balance
*We are searching for a way to buy more fruits and vegetables at the regional farmers market, because I find it horrible to have all the plastic here, when I come from ALDI. […] I see that with my friends, who all have become mothers, they really get into that thing, they celebrate it. Maybe one rather does it then, because one needs time and I think a bit routine. (20204, female, 30 years)*	Taking opportunities at turning points in life	
*R: Within Mannheim we only ride our bikes and only if we go somewhere further away we take the car […]. And in this case I think the train is not more efficient, if four persons are seated in the car […]. And the train doesn’t go everywhere. (20101, female, 43 years)*	Taking decisions based on comparison to other actions in own life or to others	

Code of participant in parenthesis.

**Table 9 ijerph-17-01297-t009:** Different perspectives on equity illustrated by the example of air travel.

Interindividual Equity	Interindividual and Intersectoral Equity
*Well, I can remember that I left [the simulation game] and found that we can’t do a lot more than we already do. We actually already do enough from our perspective (laughing) […]. If we would consume more, than it would be easier to save more. For instance, if we would fly for vacation, then it would be easier to say, I stop flying. (20101, female, 43 years)*	*To reduce flying is difficult, if one wants to go somewhere […] and flying is the only option. Of course, one can say, I don’t want to go to the US anymore due to organic reasons. But this is not the case, I do want to go to the US at some time. Thus, I would never stop flying due to CO_2_ reasons. Other people […] fly three times a week due to professional reasons and I should refrain from my holiday flight once a year? Nope. (20228, male, 45 years)*

Code of participant in parenthesis.

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
