# Peer review of "The Role of Health in Households’ Balancing Act for Lifestyles Compatible with the Paris Agreement—Qualitative Results from Mannheim, Germany"

_ijerph, 2020, doi:10.3390/ijerph17041297_

Round 1
Reviewer 1 Report
An interesting paper looking at facilitators and barriers to making environmental changes in households. It is not clear what the aim of the qualitative aspect of study is and it is recommended for this to be more specific.
It would have been helpful for this reviewer to have some information related to the characteristics of the interviewees included with the quotes. As the participants were selected for the study based on different categories (eg., children in the household, climate skeptic, income, location of household) it would be interesting for the reader to be aware of any differences in responses. While this may lead to the development of educational programs that target certain groups, it can also inform where policy changes may be more effective.
The authors need to be consistent with their referencing - for example in the discussion many of the references are written out while others are numerical. Furthermore, there are some English language issues with some of the wording used - eg., Table 6 page 13 correct wording is planes rather than plains; line 485 should be price rather than prize. One quote is in German rather than English in Table 5. It is recommended that the authors seek assistance with their English writing and grammar.
In addition, the authors have included more information related to the interviewees in Table 5 (gender and age in some cases) but not for any of the others.
Reviewer 2 Report
The manuscript “The Role of Health in Households’ Balancing Act for Lifestyles Compatible with the Paris Agreement– Qualitative Results from the German Sample of a European Mixed-Methods Study” presents an interesting and needed study about the willingness of people to change their behaviour to save GHG emissions. However, the manuscript needs revision in its presentation and writing before publication.
The study clearly only presents only results from the German sub-group, but by starting to read (starts by title) it is suggested to get more results/information about the other European sub-groups. It should made clear in the beginning that this paper is about a study of German households that was part of a larger study and not present the larger study and at some point just starting to talk about the German results.
The introduction and method part needs revision in structure, context and writing. Sometimes it reads as if someone is telling me instead of reading a scientific paper. The authors should revise the paper and condense the information, be more precise and should not repeat themselves. Specifically in the method part, it is only important what was done and what method they followed. Phrases like “it is important” should not be included in the methods.
The results are well structured and presented. The authors only need to be carful not to start the discussion at this point.
Overall, the study is interesting and worth publishing in the journal, the authors just need to take a bit more time to prepare the manuscript carful and in line with the author guidelines.
Comments
Writing: The authors should not use terms like “This is because...”
Title: Leave the part about the European study; it does not matter for the presentation of these results.
Abstract: Yes, the abstract should be structured following background, method, results and conclusion, but it should not be numbered and given by headline.
I would suggest to use IS units and use % instead of percent and the use of GHG as abbreviation is more common to use in literature if it is used through out the text and not just mentioned once or twice. The authors are very inconsistent in the use of their abbreviations and units.
L36-37 Looking at the 72%.: I am not sure about the formulation. Do the authors mean from total anthropogenic GHG emissions? Or really just from consumption? What is the percentage from total emissions?
L41-42 Name the other GHGs.
L88 – 103 I am confused why the authors present results already in the introduction. This should be moved to the method and result section accordingly.
L110-113 Isn’t that the conclusion? Main findings should be summarized in the end of a paper not in the beginning.
The method part would benefit from a more precise structure. Maybe by adding the questionnaire as a supplementary and refer to it. Or include a flow chart about the procedure in general and reduce the description.
L135 Reference or link to FCS-Tool is missing.
L137 CH4 and N2O have a higher GWP as CO2 and this is considered when all GHGs are added. Use IPCC reference and rephrase correctly.
L173 What does AH mean?
L207 Why only German households? I thought the study was about 4 European countries?
L215 check writing, space missing
L240-241 If the authors want to differentiate between interviewers they should use objective descriptions like reviewer 1 (R1) etc. instead of initials. If it is used in the following text.
Table 2 gives a good overview. Would it be possible to add the references to each of the GHG reduction measure? Or was this table published before?
(L256) okay, this reference should be given earlier.
L261 grouped them in categories and subcategories
L270 use only 1.5° and delate degree
L288 it does not matter why health category is presented first. The authors decided on a structure, now they should present and not talk about presenting.
L301 This was because... (do not use this phrasing) And this is not a result, this is discussion.
L314-352 Finally, a nice comprehensive description of the results.
L367 -380 Interesting. Do people feel they get misleading information? Or do they don’t trust the information they get? Would they feel more comfortable to change if policies are in place to promote insulation or use renewable energy?
L386-393 So were the majority of people prepared to pay higher prices to achieve sustainable options? – I read the barriers later on. Would be good to mention here again, that they just point out what they think is important without considering the consequences to their own life.
L398 Last sentence is a repeat, delate.
Table 5: Check table. Third quote: “R:”?, Fifth quote is in German.
Results: Overall, the results are well presented, structured and focused on the main outcome. The authors should only consider reducing the introduction part in the beginning of the results and be carful not to start the discussion (or change the structure to a combined result and discussion part).
L600 Is it really a strength to choose participants rather than have a big pool of participants representing the average of the population? Discussion?
L597-610 I do not see a real discussion about the method. What are limitation? Were there regrets after the analysis that there might have been questions they had not asked?
L618-635 Reads more like a result and not a discussion
The conclusion summarises the main outcomes and consequent steps nicely.
